# Joint Diagnostic Method of Tumor Tissue Based on Hyperspectral Spectral-Spatial Transfer Features

**DOI:** 10.3390/diagnostics13122002

**Published:** 2023-06-08

**Authors:** Jian Du, Chenglong Tao, Shuang Xue, Zhoufeng Zhang

**Affiliations:** 1Key Laboratory of Spectral Imaging Technology CAS, Xi’an Institute of Optics and Precision Mechanics, Chinese Academy of Sciences, Xi’an 710119, China; dujian@opt.ac.cn (J.D.); chengltao@126.com (C.T.); turboxuego@163.com (S.X.); 2Xi’an Key Laboratory for Biomedical Spectroscopy, Xi’an 710119, China

**Keywords:** hyperspectral imaging, spectral-spatial transfer feature, deep learning, tumor tissue, joint diagnosis

## Abstract

In order to improve the clinical application of hyperspectral technology in the pathological diagnosis of tumor tissue, a joint diagnostic method based on spectral-spatial transfer features was established by simulating the actual clinical diagnosis process and combining micro-hyperspectral imaging with large-scale pathological data. In view of the limited sample volume of medical hyperspectral data, a multi-data transfer model pre-trained on conventional pathology datasets was applied to the classification task of micro-hyperspectral images, to explore the differences in spectral-spatial transfer features in the wavelength of 410–900 nm between tumor tissues and normal tissues. The experimental results show that the spectral-spatial transfer convolutional neural network (SST-CNN) achieved a classification accuracy of 95.46% for the gastric cancer dataset and 95.89% for the thyroid cancer dataset, thus outperforming models trained on single conventional digital pathology and single hyperspectral data. The joint diagnostic method established based on SST-CNN can complete the interpretation of a section of data in 3 min, thus providing a new technical solution for the rapid diagnosis of pathology. This study also explored problems involving the correlation between tumor tissues and typical spectral-spatial features, as well as the efficient transformation of conventional pathological and transfer spectral-spatial features, which solidified the theoretical research on hyperspectral pathological diagnosis.

## 1. Introduction

Compared with traditional pathological diagnosis [1], automated pathological diagnosis is more objective, and thus it can provide effective aided diagnostic information and reduce the complicated workload of pathologists. At present, the field of automated diagnosis mostly focuses on digital pathology research [2]. Digital pathology is based on a large number of labeled pathology datasets for analysis and modeling, which has the advantages of low difficulty in quantitative analysis and high utilization and repeatability of information, and it has also achieved results in staging and classification of several primary tumors [3,4,5]. However, the current development of digital pathology technology has also encountered some bottlenecks [6]. For example, it is challenging to improve model performance on existing bases [7]. Moreover, the cumulative effect of more datasets is becoming weaker [8]. The relevant models are effective in primary tumor staging but difficult to apply to more complex subtyping, thus hindering large-scale clinical applications [9].

Hyperspectral imaging technology is expected to be an effective way to solve the above-mentioned problems. Spectral imaging originated in the field of traditional aerial remote sensing, and it is now crossing the broad-band and high-resolution hyperspectral imaging technology [10,11,12]. In recent years, breakthroughs in imaging methods and key parameters such as spectral resolution have made it possible for medical spectral detection at a close range [13,14]. Similar to the human fingerprint, between tumor and normal tissues (such as different tissue structures or different subtypes of the same tissue), due to their different basic components, the degrees of absorption, reflection, and scattering of light waves vary. Then different light patterns (feature peak or feature valley) are produced. This specific spectral “fingerprint” is the theoretical basis for achieving hyperspectral pathological diagnosis [15,16].

Hyperspectral technology can automatically analyze and detect regions of interest, and quantitatively evaluate the degree of variation in lesions. For example, Li used micro-hyperspectral imaging to analyze skin tissue and segmented epithelial cells based on spectral angle and spectral distance, achieving a better performance than the simple pixel-by-pixel segmentation method [17]. However, due to the lack of spatial information, it is challenging to further improve the accuracy of localization and segmentation. Akbari obtained micro-hyperspectral images of lung metastatic tumors in the range of 450–950 nm, and used support vector machines for classification, which achieved a detection sensitivity of 92.6% for lung cancer tissue [18]. However, this algorithm is only sensitive to transmission intensity and cannot recognize subtle spectral differences. Jong used hyperspectral imaging and CNNs to discriminate healthy tissue from tumor tissue in lumpectomy specimens and achieved a Matthews correlation coefficient (MCC) of 0.92 on the tissue slices, thus indicating the potential of hyperspectral imaging to classify the resection margins of lumpectomy specimens [19]. Hu established a micro-hyperspectral dataset based on 30 gastric cancer patients and extracted deep spectral-spatial features by 3D-CNN, which achieved a classification accuracy of 97.57% in the dataset of type IV undifferentiated gastric cancer [20]. However, this method was only tested on a small sample dataset of a single subtype, and its applicability and scalability in pathological slide diagnosis still need to be validated.

The above studies indicate that more and finer spectral-spatial features of tumors can be extracted through appropriate preprocessing methods and deep learning models. Hyperspectral pathological diagnosis technology is expected to become a reliable means to solve the problem of automatic rapid diagnosis. However, due to the constraint of factors, such as the method of clinical sample acquisition [18,21], performance of micro-hyperspectral equipment [22] and progressiveness of analysis algorithms [17,23], a systematic and complete theoretical research on spectral diagnosis has not been performed. moreover, problems involving the correlation between tumor tissues and typical spectral-spatial features, as well as the efficient transformation of conventional pathological and transfer spectral-spatial features, have not been effectively resolved. Clarifying the above problems is the key to hyperspectral pathological diagnostic research and the foundation for the ultimate realization of a joint diagnostic method.

At present, medical hyperspectral research is still in the early development stage. With the influx of more sample data and the establishment of multi-dimensional efficient models, medical hyperspectral technology is expected to become a powerful supplement to existing histopathological tools. This paper applies a multi-data transfer learning model to the classification task of micro-hyperspectral images to explore the differences in transfer features between tumor tissues and normal tissues. It focuses on how to use deep transfer learning models to build strong bridges between multi-source data and integrate more morphological information into the hyperspectral data model. Finally, a joint diagnosis scheme is established based on the entire process of data acquisition and analysis to achieve an efficient pathological diagnosis.

## 2. Materials and Methods

### 2.1. Experimental Framework

The experimental framework and chapter arrangement of this paper are shown in Figure 1. We used self-developed equipment (see Section 2.2 for details) to collect hyperspectral data (HS Data) and pathological color data (Color Data), including three datasets: D1, D2, and D3 (Section 2.3). All samples need to be labeled with detailed categories by doctors. D1 is divided into spectral samples (D1-S), spectral-spatial samples (D1-SS, SS for short), and image samples (D1-Color) by different preprocessing methods (Section 2.4).

In this study, we established an optimal classification model BufferNet (Section 2.5) for D1-S to extract the typical spectral features of tumor tissues. Then, a more comprehensive structure feature of tumor tissue (Section 2.6) was learned by a multi-data transfer learning model to solve the problem of insufficient hyperspectral data samples. Based on the Trans-CNN model trained on D2, D1-Color data were incorporated for further training to obtain CT-CNN-2. At the same time, we added D1-SS data to train SST-CNN-2 to extract spectral-spatial transfer features, and then we compared the models and selected the optimal transfer model.

Finally, in order to apply the proposed CNN models to actual automated pathological diagnosis, a joint diagnosis method of spectral-spatial feature was proposed by combining the preprocessing with the above optimal models. This method was applied to gastric cancer and thyroid cancer data (D3) to verify the overall system performance, and was also integrated into software for clinical practice (Section 4).

### 2.2. Micro-Hyperspectral Imaging System

The data-acquisition equipment used in this study is a self-developed micro-hyperspectral imaging system MICROspecim, as shown in Figure 2a. The system adopts the principle of built-in scanning and dispersion, and it mainly consists of a scanning spectral imaging system, microscopic imaging system, and data acquisition system. During data acquisition, the projection beam enters the scanning spectral imager through the sliced sample and passes through the built-in slit, collimator, dispersion element and focusing lens, in sequence, finally obtaining one-dimensional spatial information and spectral information of the target on the area array detector. The data-acquisition system stores real-time images and sends instructions to control the movement of the precision displacement table to obtain other dimensional spatial information of the target. At the end of the scan, a data cube (Figure 2b) can be obtained. The format of the obtained data is 256(*λ*) × 1000(*x*) × 1000(*y*), where 1000 × 1000 is the image size, and 256 is the number of spectral channels. In subsequent applications, the first 30 and last 26 bands are removed, and the visible and near-infrared spectral data of 410–900 nm are retained. The data-storage format adopts the BIP (Band Interleaved by Pixel), which is stored in pixel-band order, and the storage process takes about 5 s. The spectral resolution and spatial resolution reached 3 nm and 0.5 µm respectively. Compared with the previous acquisition equipment [20], the equipment used in this study has a larger image size and faster scanning speed. The larger image size means a larger field of covered view and more information in a single scan. The shorter scanning time means faster and easier access to valuable information, which is more conducive for clinical applications.

### 2.3. Experimental Dataset

The experimental dataset of this study consists of hyperspectral data and pathological color data, as shown in Table 1. The hyperspectral data samples of D1 and D3 were all from the Department of Pathology, the First Affiliated Hospital of Xi’an Jiaotong University. They were collected by MICROspecim and included 50 cases of gastric cancer and 37 cases of thyroid cancer pathology. While preserving the original hyperspectral data, pseudo-color images were also saved. Then, the doctors labeled them, mainly marking the boundaries of tumor tissue and normal tissue, as well as the specific location of partial cancer and normal cells, and finally forming labeling documents. The pathology color data of D2 were collected by a high-definition camera, including various mixed adenocarcinoma samples (gastric cancer, thyroid cancer, etc.), which also contained detailed labeling documents. Afterward, hyperspectral data needed to be preprocessed. The sample extraction process is shown in Image (Figure 2d)/Spectral-spatial (Figure 2e) samples are extracted according to the label information at a size of 250 × 250. For normal samples, the entire area is traversed for extraction, while samples with too much of a blank area are removed. Spectral samples (Figure 2c) are extracted based on the manually labeled pixel information of cancer cells and normal cells. After standardization, D1-S has a total of 26,293 spectral samples (size of 200 × 9 × 9). After principal component processing, D1-SS of spectral-spatial samples (size of 3 × 250 × 250) and D1-Color image samples (size of 3 × 250 × 250) each have 4109 samples. The overall experimental dataset includes 40,957 image/SS samples and 42,618 spectral samples, as shown in Table 1 (see for more details).

### 2.4. Hyperspectral Data Preprocessing

#### 2.4.1. Data Standardization

To ensure the spectral consistency of different tissue samples, a standardized data calibration process is required. Based on the conventional calibration method of remote-sensing images, the spectral calibration operation under transmission spectrum condition is added. The main steps are as follows (as shown in Figure 3):1.Whiteboard correction: Firstly, the hardware system of the micro-hyperspectral imaging equipment is corrected by using a white board as the reference target to obtain the ${MD}^{raw}$. Correction parameters are built into the acquisition software system and used to eliminate hardware differences, including the focal plane.2.Flat-field calibration: For the uneven brightness of the same sample caused by uneven smear or different coloring degrees of colorant, real-time calibration is performed during the acquisition process to obtain the ${MD}^{even}$. For each column during column scanning, real-time averaging is performed, and then the average is divided by each pixel value in this scanning column. As shown in Equation (1), ${MD}_{r,c}^{raw}$ and ${MD}_{r,c}^{even}$ are the spectral curves of MD before and after flat-field calibration at position (r,c), respectively. To avoid the influence of outliers, the maximum and minimum 150 spectral values in each column are excluded. Moreover, p is the remaining number of pixels.
(1)$${MD}_{r,c}^{even}=\frac{{MD}_{r,c}^{raw}}{\frac{1}{p}\times \sum_{k}^{p}{MD}_{k,c}^{raw}}$$3.Transmission spectrum standardization: Due to the slight differences in slice thickness and light source intensity among different samples, the overall image brightness may vary. As shown in Equation (2), the standardized transmission spectrum ${MD}^{tran}$ is obtained by dividing ${MD}^{even}$ by SC, and the average spectrum SC of the part without medium coating or background region is selected as the reference.
(2)$${MD}_{r,c}^{tran}=\frac{{MD}_{r,c}^{evev}}{SC}$$

#### 2.4.2. Principal Components Analysis

The principal component analysis (PCA) is a multispectral orthogonal linear transformation based on statistical features [24]. Its basic idea is to recombine the data of n correlated spectral bands into relatively fewer spectral principal components, that is, reducing multiple spectral dimensions to a linear combination of spectral principal components and using the extracted principal components to approximately represent original multidimensional spectral data [25]. The specific calculation process is as follows:1.Subtract the mean value of each feature (data need to be standardized).2.Calculate the covariance matrix of samples x and y.
(3)$$Cov(x,y)=\frac{\sum_{i=1}^{n}\left(x_{i}-\bar{x}\right)\left(y_{i}-\bar{y}\right)}{\left(n-1\right)}$$3.Calculate the eigenvalues and eigenvectors of the covariance matrix. If the covariance is positive, x and y are positively correlated. If it is negative, x and y are negatively correlated, and if it is 0, x and y are independent. If Ax=λx, then λ is the eigenvalue of A, and x is the corresponding eigenvector.4.Sort the eigenvalues in descending order, select the top m eigenvectors, and transform the original data into a new space constructed by m eigenvectors.

Through PCA transformation, multiple high-dimensional spectral features are concentrated into a minimal number of principal components, reducing data redundancy and computational complexity. The resulting principal components are also uncorrelated with each other.

#### 2.4.3. Data Promotion

To make the most of limited training data, we adopted data promotion to increase the number of training samples. Since the microscopic structural features of tumor tissue have important morphological significance, simple operations such as stretching, compression, magnification, and reduction cannot be performed. For the extracted image samples, we used translation, flipping, and rotation to augment training samples, as these can effectively suppress overfitting and improve the generalization performance of the model.

### 2.5. BufferNet Model

Micro-hyperspectral data contain three dimensions: spectrum, width, and height. The spatial information contained in the width and height dimensions is closely related to the spatial information in the spectral dimension. Extracting features from a hyperspectral medical image (HMI) without losing its correlation is extremely important. Therefore, a CNN-based modeling approach was chosen to achieve fine spectral feature extraction for microscopic tumor tissue [26]. CNN is a feedforward neural network of deep learning which is widely used in image processing and computer vision. Compared with traditional machine-learning algorithms, CNN has better feature-extraction and image-classification capabilities due to operations such as convolution, weight sharing, and pooling.

Since 1D-CNN filters can only extract features of one-dimensional spectral curves and 2D-CNN extract spatial features of two-dimensional images, they cannot extract spectral-spatial features of HMIs [27,28]. In contrast, 3D-CNN is more suitable for extracting features from data cubes by sliding convolution kernels in three dimensions, including two spatial and one spectral direction [29]. For a data cube, $D\in R^{S\times H\times W}$, a 3D-CNN performs convolutional operation with a 3D filter, $F\in R^{K\times I\times J}$, to obtain a 3D feature map, $M\in R^{U\times V\times Z}$. Assuming the stride is t, the value of M[s,h,w] at (s,h,w) can be calculated according to Equation (4), in which ∗ represents convolutional operation. Image F is traversed and finally outputted M. Since D is not decomposed or averaged, the correlation between the spectral and spatial information is preserved.
(4)$$M\left[s,h,w\right]=\left(D*F\right)\left[s\times t,h\times t,w\times t\right]=\sum_{k=1}^{K}\sum_{i=1}^{I}\sum_{j=1}^{J}D[s\times t-k,h\times t-i,w\times t-j]\times F[k,i,j]$$
where s, h, and w represent the coordinates in the three output directions, namely the position of length, width, and spectral dimensions; and I, J, and K represent the offset of the convolution kernel in these three directions, which collectively determine the receptive field size of the layer. For cell objects in the data cube, after multiple down-samplings, the resolution of feature map will be smaller than 1 pixel in the feature map. Therefore, we designed two types of convolutional layers, i.e., a down-sampling layer and buffer layer, and this network model is named BufferNet. To expand the receptive field of neurons, the stride of down-sampling was set to 2. The stride of buffer layer was fixed at 1 to enhance the representation without reducing the resolution of the feature map. The convolution kernel is 3 × 3 × 3, which is used to extract detailed information of the data cube. The network structure of BufferNet is shown in Figure 4; it consists of three buffer blocks, and each block consists of 1 down-sampling layer and 3 buffering layers. To speed up the training process, each 3D convolutional layer is followed by a 3D batch normalization layer.

### 2.6. VGG-16 Model

Transfer learning [30,31] allows us to apply a pre-trained neural network model to a specific dataset. This means that when we solve the current tumor classification problem, we can start from a model trained on similar medical diagnostic problems, instead of training a new model from scratch. For the practical task where hyperspectral data were difficult to obtain in this study, transfer learning can be used to apply the relationships learned from previous models on other pathological data to this field [32,33]. VGGNet [34] is a relatively mature pre-trained model, which is usually slightly improved based on actual needs and then transferred to specific tasks in order to fully utilize existing data resources and save time costs. The deep features extracted from these pre-trained networks usually have a good generalization performance. For example, Sermanet [35] and Kermany [36] applied pre-trained models to classification tasks of mouse brain images and eye images, respectively, and achieved high accuracy.

VGG-16 was used as the pre-trained model to extract the spectral-spatial transfer features in this paper. The network structure of VGG-16 is shown in Figure 5. The features of the model itself are sufficiently generalized, and there is no need to modify too many weights. Therefore, the structure and pre-trained weights of the pre-trained model were retained as initialization parameters, and the part above the fully connected layer was discarded for learning new weights. The entire network uses the same size for the convolution kernel and pooling kernel, and this is conducive to extracting features of image detail and provides good scalability. During the training process, fine-tuning and freezing operations are adopted, and all network layers are initialized with pre-trained weights. Because the entire network has a huge entropy capacity, we freeze the lower-level convolution (learning general features) and only fine-tune the later convolutional layers (learning more specific features).

## 3. Results

### 3.1. Classification Results of Typical Spectral Features

In training, the batch size of BufferNet is set to 128, the learning rate is set to 0.0001, the momentum is set to 0.9, and the number of iterations is 30. Stochastic gradient descent is used as the optimizer. The training samples are shuffled at each iteration. In the comparison experiments, we chose the previously designed model, SS-CNN-3 [20], and the classical method, 3D-ResNet, to compare with BufferNet. SS-CNN-3 was the previous optimal model; it achieved a better performance than conventional machine learning and 2D CNN methods. ResNet was originally a network structure based on 2D convolution [37], which achieved an excellent performance in the ImageNet classification task, effectively solving the problem of gradient vanishing in deep neural networks. To facilitate the comparison of network architecture performance, we replaced 2D convolution kernels with 3D ones to build a 3D-ResNet, and the input sizes of these models were uniformly set to 200 × 9 × 9.

Table 2 shows the classification performance of three different algorithms on the test dataset. The results show that BufferNet is the best algorithm, with an overall accuracy of 96.17%, which is higher than the 95.83% for 3D-ResNet and 95.20% for SS-CNN-3. Since these three models mentioned above all use 3D convolution as the filter, differences in performance can only be caused by differences in model structure. This indicates that the buffering layer plays an important role in improving the extraction ability of spectral features. Compared with 3D-ResNet, BufferNet achieves better results, and its model is lighter and has fewer parameters. The training and testing time is relatively less, making it more conducive to practical application.

### 3.2. Results of Spectral-Spatial Transfer Features

In order to fully utilize more samples from cancer datasets, it is necessary to extract more generalized features of the tumor tissue structure and then train our specific hyperspectral cancer data on this basis. The construction process of the spectral-spatial transfer model is shown in Figure 6. VGG-16 is used as the pre-trained model, and Trans-CNN is trained based on the pathology color dataset D2. Then Trans-CNN is used as the pre-trained model for subsequent processing, extracting transfer features corresponding to conventional pathology and hyper-spectrum from the hyperspectral gastric cancer image dataset D1-C and spectral-spatial dataset D1-SS, respectively.

#### 3.2.1. Transfer of Conventional Pathology

The training process and results of conventional pathological transfer model are as follows:Build the VGG-16 model and load its weights. Use the training and testing data of D2 as input to run the VGG-16 model. Then build a fully connected network to train the classification model Trans-CNN.Based on the Trans-CNN model, use the weights of each layer as initialized parameters, and use the hyperspectral gastric cancer dataset D1-C as the model input to extract spatial transfer features. Train a completely new classification model, CT-CNN-1, with a low learning rate. Figure 7a shows the training curve of CT-CNN-1, and the accuracy reaches 91.53% after 50 iterations.Fine-tuning is performed on the basis of CT-CNN-1 to further improve the model’s performance. Freeze all convolutional layers (Blocks 1–4, shown in Figure 5) before the last convolutional block, and only train the remaining layers (Block 5 and FC) to obtain the classification model CT-CNN-2. The SGD optimization method is used for training, with a learning rate of 0.0008, batch size of 50, and epoch of 17. The training curve is shown in Figure 7b.

CT-CNN-2 achieved a classification accuracy of 92.20% after 17 iterations. The results show that the pre-training on the pathological big dataset D2 played a positive role in improving the classification performance of the model, and moderate fine-tuning was also conducive to improving the model performance.

#### 3.2.2. Transfer of Spectral-Spatial Data

D1-SS is replaced as the model input to extract spectral-spatial transfer features and train SST-CNN-2. This part of the training process is the same as Section 3.2.1, except for the different input dataset. First, a PCA is performed on all data in the training and testing sets. Figure 8 shows the results of the hyperspectral gastric cancer sample after the principal component analysis, including grayscale images of different principal components and pseudo-color images.

Figure 9 summarizes the amount of information contained in each principal component. It can be seen that the first 10 principal components contain over 98.2% of spectral information. Among them, the first principal component, PC1, contains 76.2% of the information, while the information content is relatively low after the fifth principal component. Therefore, the first five principal components are mainly used. We synthesized pseudo-color images with different principal components to form the D1-SS dataset, including PCA123, PCA134, PCA145, etc., to replace the hyperspectral gastric cancer dataset D1-C as input. At the same time, in order to better compare the effect of PCA processing, each principal component (PC1–PC5) is used as input to train its respective models.

The final results are shown in Table 3. It can be seen that when we use three principal components, the accuracy from high to low is: PCA134 (95.46%) > PCA514 (94.39%) > PCA145 (94.25%) > PCA123 (80.78%). PCA134 provides the best performance, while PCA123 provides a poorer performance.

### 3.3. Results of Joint Classification Diagnosis

In the above, classification models of gastric cancer tissue were established from the perspectives of spectral features and spectral-spatial transfer features. Different classification models have different characteristics. In this section, we combine the above optimal models, make use of their respective advantages, and refer to practical application experience to establish a joint diagnostic method, which includes the steps shown in Figure 10.

Hyperspectral data acquisition of pathological sections: Following the method described in Section 2.2, hyperspectral images of 20× sections are acquired. According to the imaging size of equipment and sample radius, in order to cover all sample tissues on the section, an average of 6–8 images per section need to be collected.Classification of spectral-spatial samples: The size of each original hyperspectral image is 256 × 1000 × 1000. The SS samples are extracted and preprocessed following the method in Section 2.3 and Section 2.4. Each hyperspectral image can generate 16 small SS samples, which are then classified based on the spectral-spatial information separately. The trained model SST-CNN-2 is applied. The numbers for the small samples diagnosed as normal tissue are recorded. For the samples diagnosed as cancerous tissue, further processing is carried out.Spectral information extraction of cancerous tissue: For samples diagnosed as cancerous tissue by SST-CNN-2, spectral angle matching and unsupervised clustering methods are used to remove red blood cells, lymphocytes, cytoplasm, and interstitium from the sample data, leaving only gastric cells and cancer cells as classification samples for the next step.Spectral classification of cancerous tissue: The trained model BufferNet is applied to classify cancerous tissue and normal tissue.According to the spectral classification results from Step 4 and the spectral-spatial classification results from Step 2, the joint classification probabilities of each small sample belonging to the cancerous sample are assigned to determine the final category of the original hyperspectral data.

Taking the gastric cancer sample in Figure 11 as an example, the collected hyperspectral data are divided into 16 small SS samples (Nos. 1–16, Figure 11b) with a consistent size. Figure 11a shows the result of the SST-CNN-2 classification model in which 11 small samples are diagnosed as gastric cancer (>0.9) and 2 as normal tissues (<0.1). These 13 small samples were determined and do not require further processing. In addition, the probabilities of gastric cancer tissue in No. 4, No. 8, and No. 15 are 0.826, 0.142, and 0.730, respectively, and further processing is still required to determine their categories. Figure 11b shows the result of spectral classification after passing through BufferNet. Each suspected cancerous pixel is classified, and the red marking indicates that the sample point is diagnosed as gastric cancer. It can be seen that, except for No. 12 and No. 16, other regions are more or less distributed with cancerous sample points, and this is similar to the classification results obtained in Figure 11a. The results of Figure 11a,b are combined to obtain the final joint classification result (Figure 11c), in which the categories of No. 4 and No. 15 are further confirmed after spectral classification. In practical applications, if there is a cancerous probability of 0.377, like in No. 8, in order to obtain a higher sensitivity indicator, we usually diagnose this sample as a cancerous area. By extending the above process to the entire section, we can obtain the category of each hyperspectral image on the section and determine the cancerous probability of this section.

In order to have a clearer understanding of this diagnostic process, we calculated the probability of each pixel being classified as gastric cancer based on the spectral-spatial joint diagnosis results, as shown in Figure 11d. The closer the color is to red, the higher the probability that the area belongs to a cancerous area, and the closer the color is to blue, the higher the probability that it belongs to a normal area. Comparing with the true area marked by the doctor in Figure 11e, it can be seen that our classification results are quite accurate. Figure 12 shows the classification results of a gastric cancer tissue and a normal tissue. The entire region in Figure 12a is undifferentiated adenocarcinoma, and Figure 12c is normal mucosal tissue. Moreover, the distribution of each tissue is clearly shown in the corresponding classification results (Figure 12b,d).

## 4. Discussion

Previous studies have shown the effectiveness of spectral-spatial features in spectral diagnose. For example, Sun collected 880 scenes of multidimensional hyperspectral cholangiocarcinoma images and realized further diagnose based on the features from patch prediction [38]. Liu found that the spectral data of nuclear compartments contribute more to the accurate staging of squamous cell carcinoma compared with peripheral regions [39]. Martinez-Vega evaluated different combinations of hyperspectral preprocessing steps in three HSI databases of colorectal, esophagogastric, and brain cancers [40], and he found that the choice of preprocessing method affects the performance of tumor identification, and this partly inspired our later data accumulation. However, problems involving the correlation between tumor tissues and typical spectral-spatial features [13,41], as well as the efficient transformation of conventional pathological and transfer spectral-spatial features [15,42], have not been effectively resolved. In response to the above problems, the solution proposed in this paper has the following characteristics: (i) High-resolution micro-hyperspectral imaging combined with deep-learning models was used to extract microscopic spectral-spatial feature of tumor tissue for a precise diagnosis. Different from conventional digital pathology techniques that extract morphological information based on large data sets, micro-hyperspectral imaging focuses more on spectral differences between tissues. This approach aims to obtain the 3D data cube information of different tissues on a relatively small data set. Considering the diverse pathological malignancy and staging in gastric cancer, this research focused on extracting spectral features of pathological tissue under high-resolution conditions. Finally, accurate classification and identification of cancerous and normal tissue can be realized based on the differences of spectral characteristics. (ii) A spectral-spatial transfer model between digital pathology and hyperspectral imaging was constructed to extract comprehensive detailed features of tumor tissue, to maximize the use of existing pathology datasets. Different from previous hyperspectral pathology studies, this study was not limited to differences of spectral dimension; instead, it focused more on the learning of spectral-spatial transfer features of tumor tissue. Whether it is from the perspective of medical hyperspectral data acquisition or deep-learning model training, we are required to not be limited to the existing hyperspectral datasets but to utilize much more relevant data information and the characteristics of efficient existing models to achieve the effective fusion of transfer features of the spectral dimension and image dimension [43].

In this study, compared with the conventional pathological model, CT-CNN-2, which achieved a classification accuracy of 92.20%, the optimal result of SST-CNN-2 was improved by 3.26%. This indicates that the generalized tumor structure features obtained from pre-training have a positive effect on the overall model classification performance. The spectral-spatial transfer feature extracted based on SST-CNN-2 contains more tumor details, so the performance of the spectral-spatial model is significantly better than that of the conventional pathological model. From the results of the spectral-spatial model, we can see that PCA134 provides the best performance, while PCA123 provides a poorer performance. The results of PCA145 and PCA514 are similar, indicating that the order of pseudo-color synthesis for each principal component has little effect on the result. When we only consider the single principal component, PC1 has the best performance, and PC2 has the worst. The reason for the poor performance of PC2 is that it mainly contains brightness information from micro-hyperspectral data, and brightness differences have a negative impact on the results. There is little difference in accuracy between the principal components after PC4. This is also consistent with the data results of the amount of information, indicating a positive correlation between the classification performance and the amount of information contained in the data.

In order to verify the effectiveness of the proposed spectral-spatial transfer model and the portability of the joint diagnostic method, we extended the joint method to hyperspectral data of thyroid cancer. Thyroid cancer is one of the most common malignant tumors of the head and neck, in which the papillary carcinoma with low malignancy is the most common. We collected a total of 5070 SS samples and 16,325 spectral samples of thyroid tissue to form dataset D3. The data preprocessing and model training process are the same as those of the gastric cancer classification model, and a classification accuracy of 95.89% is achieved finally. This accuracy is slightly higher than that of gastric cancer, which may be related to the relatively single type of thyroid cancer, less invasion, and clearer boundaries between cancer and non-cancer tissues.

This is shown in Figure 13, taking a thyroid cancer sample as an example. Figure 13a shows the result of the joint classification model, and Figure 13b shows the cancerous area marked by a doctor. In order to make the display in software faster and more intuitive, the process of result visualizing in the previous section is further improved. We use the pseudo-color image of the sample as the background, and mark the high-probability cancerous areas predicted by the model with different colors. Normal areas diagnosed are marked in green, cancerous areas are marked in blue, and suspected areas near the threshold are marked in yellow. The green, blue, and yellow dots in Figure 13c indicate the color markings for normal, cancerous, and suspected areas, respectively. The visualization effect is shown in Figure 13c, and it is highly consistent with the areas marked by a doctor in Figure 13b, indicating that the joint diagnostic method proposed in this paper is applicable to several types of cancer. In order to better apply it to clinical practice, we integrated the joint diagnostic system into the acquisition software. After data acquisition is completed, the corresponding tumor type can be selected for joint diagnostic analysis, and the final diagnosis result, probability, and visualization image are immediately presented on the software interface, as shown in Figure 13d.

In actual clinical diagnosis, doctors usually pay more attention to the sensitivity indicator, because a higher sensitivity means a lower probability of missed diagnosis. From the perspective of aided diagnosis, we hope to distinguish all suspected cancer samples. Then, we can hand them over to doctors for further processing, so as not to miss any suspected cases. Therefore, we adopt a more conservative strategy in setting the thresholds of partial models in practical application in order to obtain a higher sensitivity indicator. At present, the joint diagnostic method can complete the interpretation of a section of data in 3 min. The next step is to improve the joint diagnostic method on a larger sample base from different sources, so that 80%–90% of normal samples can be eliminated in the first stage of initial screening in the automated diagnosis application of multiple diseases. Then, only the remaining problem specimens and key areas need to be observed by doctors, and this is expected to further enhance the overall diagnostic efficiency. On the other hand, histopathological diagnosis remains a crucial step currently. We aim to establish clearer spectrum pathology correlations through further research of hyperspectral diagnosis, which can better provide medical interpretation of model results.

## 5. Conclusions

The main contribution of this study is to propose a new idea of hyperspectral pathological diagnosis, which makes full use of the fine hyperspectral features and the characteristics of large-scale conventional pathological data by combining micro-hyperspectral technology with pathological images. Based on the effective integration of the spectral and spatial transfer features of tumor tissues, the spectral-spatial transfer model SST-CCN was established. In this work, a hyperspectral database was established, including 50 gastric cancer patients and 37 thyroid cancer patients, with a total of 40,957 image samples and 42,618 spectral samples. The spectral-spatial transfer model achieved a classification accuracy of 95.46% and 95.89% in gastric cancer and thyroid cancer datasets, respectively. The results are better than the models trained on single conventional digital pathology and single hyperspectral data, thus indicating the effectiveness of hyperspectral spectral-spatial transfer features for tumor tissue classification. Combined with standardized preprocessing and classification models, a joint classification diagnosis method based on SST-CCN was established. It was integrated into acquisition software and successfully applied to clinical sample data, indicating the portability of the joint diagnostic method for different cancer tissues. The introduction of this method is not only a new exploration of pathological automation diagnosis, but also a powerful supplement to traditional histopathology.

## Figures and Tables

**Figure 1 diagnostics-13-02002-f001:**
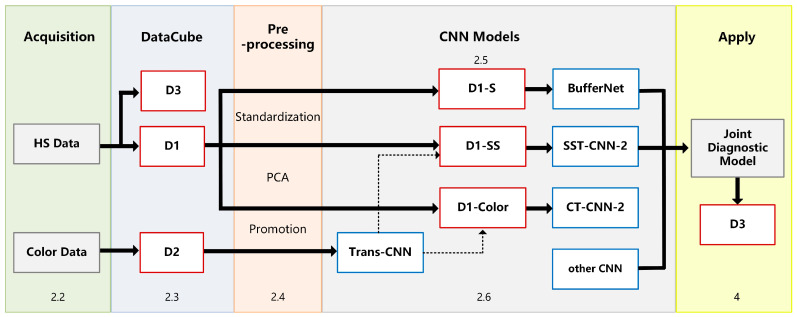
Experimental framework.

**Figure 2 diagnostics-13-02002-f002:**
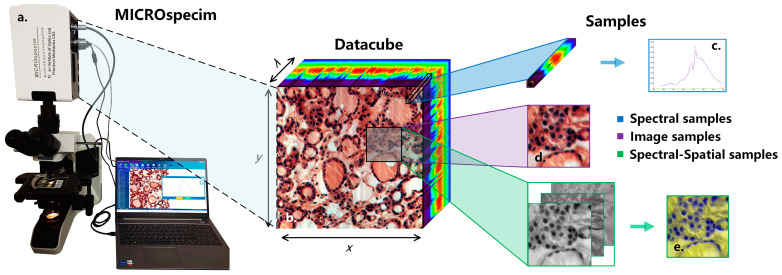
Schematic diagram of micro-hyperspectral imaging system and sample extraction process. (**a**) Micro-hyperspectral imaging system MICROspecim. (**b**) Hyperspectral datacube. (**c**) Spectral sample. (**d**) Image sample. (**e**) Spectral-spatial sample.

**Figure 3 diagnostics-13-02002-f003:**
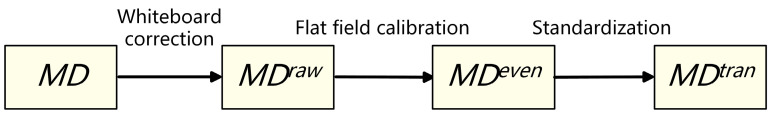
Schematic diagram of spectral calibration process.

**Figure 4 diagnostics-13-02002-f004:**
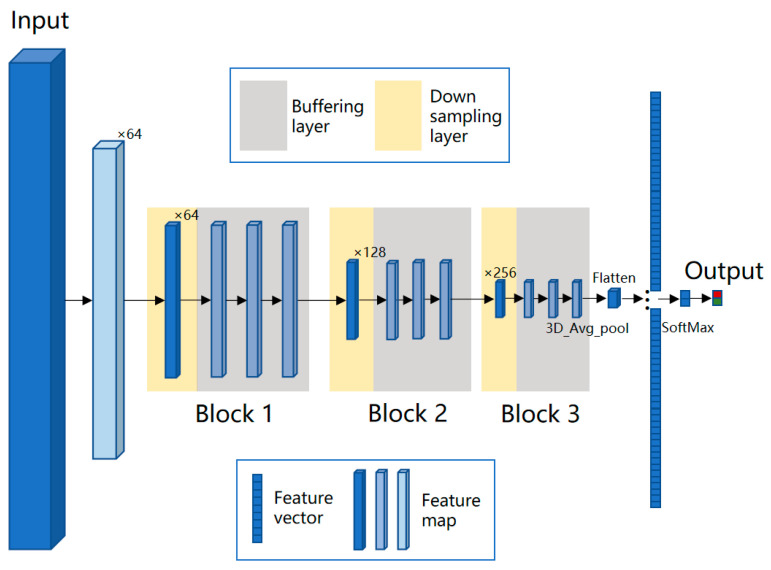
Network structure of BufferNet. BufferNet contains 13 3D convolutional layers (1 + 3 down-sampling layers; 3 × 3 buffering layers), one fully connected layer, and one SoftMax layer. The model takes spectral data (200 × 9 × 9) centered on the sample point as input. Firstly, the first filter is set to a stride of (2, 1, 1) and a kernel of (5, 1, 2) to down-sample the spectral dimension and compress it to 64 feature maps of 98 × 9 × 9. Then, the feature maps are input into the three buffer blocks, with output sizes of 49 × 5 × 5, 25 × 3 × 3, and 13 × 2 × 2, and the numbers of the output feature map are 64, 128, and 256, respectively. The output feature map of the third block is 6 × 1 × 1 after 3D average pooling and is then fed into the fully connected layer, which finally outputs two probabilities of cancer and non-cancer.

**Figure 5 diagnostics-13-02002-f005:**
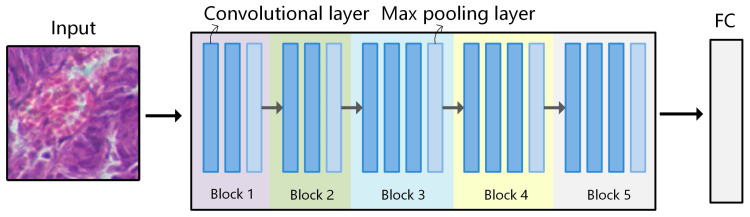
Network structure of VGG-16. VGG-16 consists of 5 convolutional blocks, where the dark blue squares represent convolutional layers, and the light blue squares represent the max pooling layers. There are a total of 13 convolutional layers (3 × 3), 5 max pooling layers (2 × 2), and 3 fully connected layers.

**Figure 6 diagnostics-13-02002-f006:**
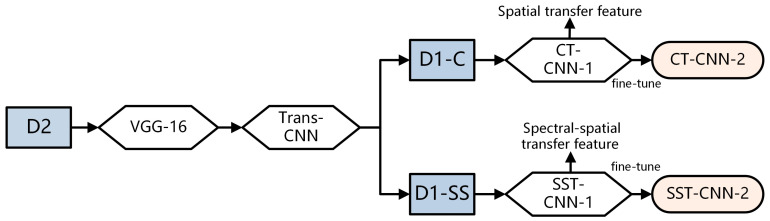
Construction process of spectral-spatial transfer model.

**Figure 7 diagnostics-13-02002-f007:**
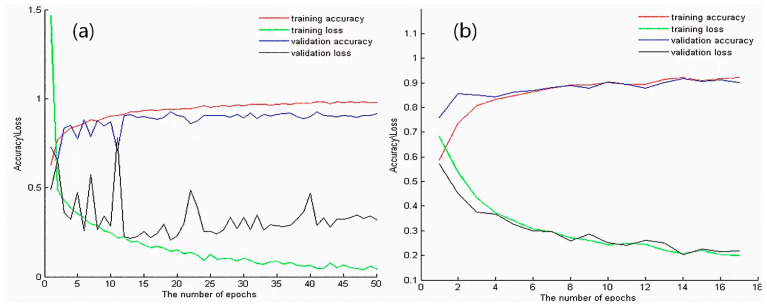
(**a**) Training curve of CT-CNN-1. (**b**) Training curve of CT-CNN-2. The red, green, blue, and black lines represent training accuracy, training loss, validation accuracy, and validation loss, respectively.

**Figure 8 diagnostics-13-02002-f008:**
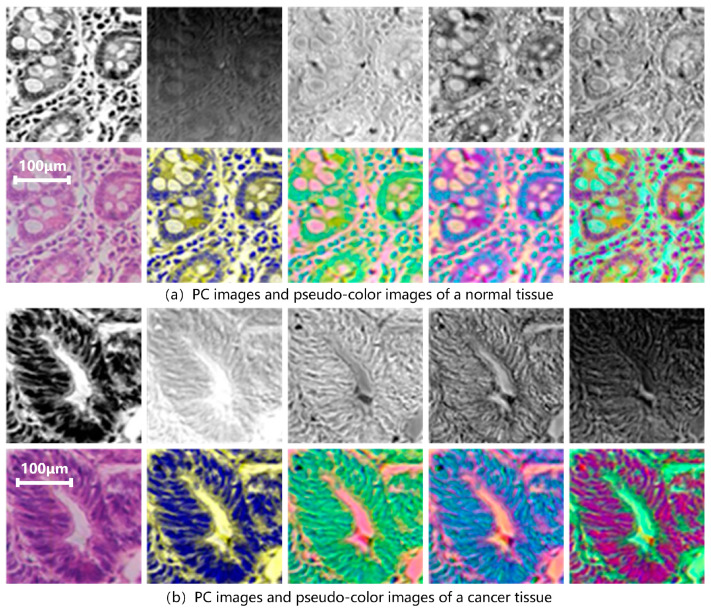
PC images and pseudo-color images of hyperspectral gastric (**a**) normal tissue and (**b**) cancer tissue. The upper row from left to right is PC1–PC5 and the bottom row is color image, PC134, PC145, PC514, PC135, in sequence. PC1 represents the first principal component. PCA134 uses PC1, PC3, and PC4 as the three channels to form pseudo-color data, and PCA135, PCA514, etc., can be obtained in the same way.

**Figure 9 diagnostics-13-02002-f009:**
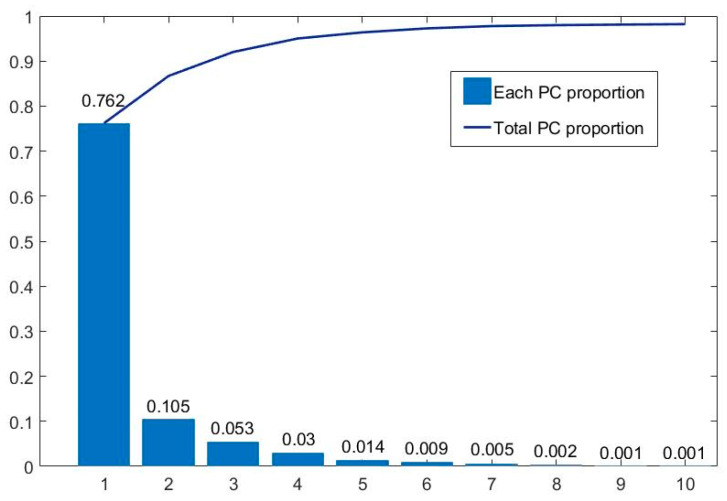
Information proportion of each principal component of hyperspectral gastric cancer data. The blue squares and blue line represent each PC proportion and total PC proportion respectively.

**Figure 10 diagnostics-13-02002-f010:**
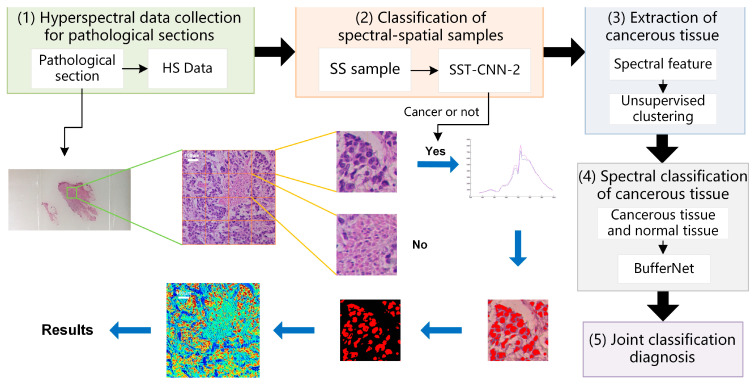
Schematic diagram of spectra-spatial joint diagnosis.

**Figure 11 diagnostics-13-02002-f011:**
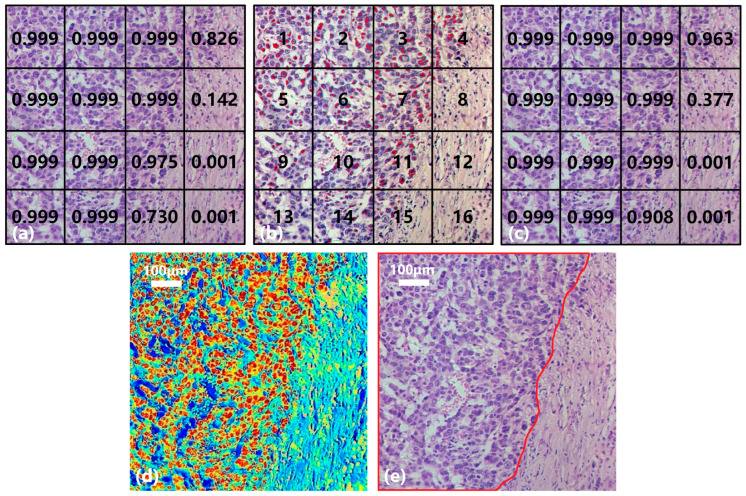
Joint diagnosis process and results of gastric cancer tissue. (**a**) SS classification result. (**b**) Spectral classification result. (**c**) Joint classification result. (**d**) Result visualization. (**e**) Mark of cancerous area.

**Figure 12 diagnostics-13-02002-f012:**
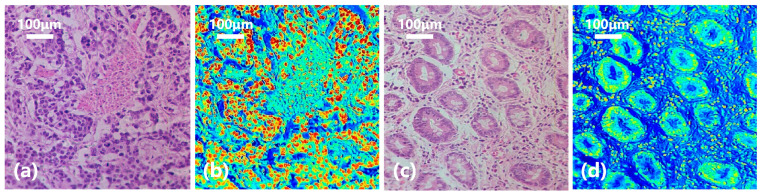
Diagnostic results of gastric cancer tissue and normal tissue. (**a**) Undifferentiated adenocarcinoma tissue. (**b**) Classification result. (**c**) Normal mucosal tissue. (**d**) Classification result.

**Figure 13 diagnostics-13-02002-f013:**
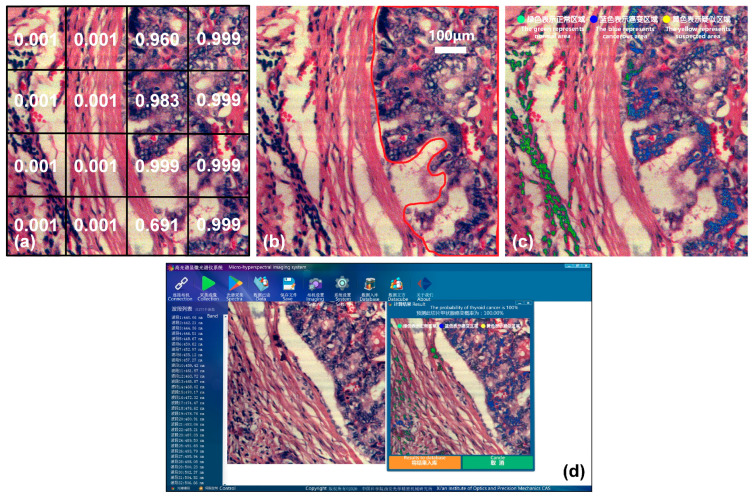
Joint diagnostic results of thyroid cancer and visualization on the software interface. (**a**) Joint classification result. (**b**) Mark of cancerous area. (**c**) Software visualization. The text after the green, blue, and yellow dots indicates normal, cancerous, and suspected areas, respectively. (**d**) Software interface of joint diagnostic system.

**Table 1 diagnostics-13-02002-t001:** Experimental dataset.

Dataset	Type	Image/SS Sample		Spectral Sample	
		Cancer	Normal	Cancer	Normal
D1	Hyper Gastric	1270	2839	11,365	14,928
D2	Mixed Pathology	9024	22,754	-	-
D3	Hyper Thyroid	1884	3186	8562	7763

**Table 2 diagnostics-13-02002-t002:** Classification results of each model.

Model	Total Parameters /Million	Training Time/s	Testing Time/s	Accuracy
SS-CNN-3	0.198	1197.16 s	1.94 s	95.20%
3D-ResNet	33.15	170,973.52 s	232.80 s	95.83%
BufferNet	8.19	48,594.21 s	57.59 s	96.17%

**Table 3 diagnostics-13-02002-t003:** Classification results of each model input.

Input	PCA123	PCA134	PCA145	PCA514	PC1	PC2	PC3	PC4	PC5
Accuracy/%	80.78	95.46	94.23	94.39	93.13	73.97	83.17	81.46	81.10

## Data Availability

Data will be made available by the corresponding author upon reasonable request.

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
