# Peer review of "Joint Diagnostic Method of Tumor Tissue Based on Hyperspectral Spectral-Spatial Transfer Features"

_diagnostics, 2023, doi:10.3390/diagnostics13122002_

Round 1

Reviewer 1 Report

In this study Du et al. developed a novel automated diagnostic system of cancer. This system has the potential to enable accurate cancer diagnosis and will become a powerful tool in clinical settings. However, there are several concerns, and the authors are requested to address the following comments.

1. A list of abbreviations for SST-CNN, MCC, HS, etc. should be provided.

2. Fig. 6, 7, 8: On what basis is it recognized as cancer cells and marked with a red dot? Is nuclear atypia the most important determinant? The red dots appear to correspond to nuclei.

3. Fig. 8: The normal tissue in (c) and (d) only contains muscle tissue. Sections containing glandular tissue should be presented as a control.

4. Fig. 9: The meanings of the colored dots should be mentioned.

Reviewer 2 Report

The paper is interesting and well written, the described approaches will surely quick diagnosis in cancer, however some issues should be adjusted:

- for this reviewer it is important to specify that the histology step will represent a key issue also requiring time for its perfect execution as well as results interpretation, that sometimes requires  an elevated know-how, maybe difficult to traslate in a "automatic" approach. 

- the initial 4 line of the discussion must be deleted

- some recent publications on the topic can be added and discussed
